# Efficacy of Core Stability in Non-Specific Chronic Low Back Pain

**DOI:** 10.3390/jfmk6020037

**Published:** 2021-04-22

**Authors:** Antonio Frizziero, Giacomo Pellizzon, Filippo Vittadini, Davide Bigliardi, Cosimo Costantino

**Affiliations:** 1Department of Medicine and Surgery, University of Parma, 43126 Parma, Italy; davide.bigliardi@unipr.it (D.B.); cosimo.costantino@unipr.it (C.C.); 2Centro di Medicina Spa, 31020 Villorba, Italy; giacomo.pellizzon@hotmail.it; 3Department of Physical and Rehabilitation Medicine, Casa di Cura Policlinico S. Marco, 30100 Venice, Italy; filippo.vittadini@gmail.com

**Keywords:** core stability, core strengthening, chronic low back pain, therapeutic exercises

## Abstract

(1) Background: Management of chronic low back pain (cLBP) is often multidisciplinary, involving a combination of treatments, including therapeutic exercises. Core stability exercises aim to improve pain and disability in cLBP increasing spinal stability, neuromuscular control, and preventing shear force that causes injury to the lumbar spine. The purpose of this study was to review the available evidence about the effectiveness in reducing pain and improving disability of core stability exercises for non-specific cLBP. (2) Methods: We perform a systematic research on common Medline databases: PubMed, Pedro, and Cochrane Library. Search results were limited to articles written in English and published between January 2005 and November 2020.The search provided a total of 420 articles. Forty-nine articles met the inclusion criteria and 371 articles were excluded. (3) Results: Core stability provides great therapeutic effects in patients with non-specific chronic low back pain reducing pain intensity, functional disability, and improving quality of life, core muscle activation, and thickness. Evidences suggest that core stability is more effective than rest or no/minimal intervention and combination with other types of exercise for cLBP have shown grater efficacy. (4) Conclusion: Core stability could be proposed in a comprehensive approach in cLBP, the combination with other modalities of therapeutic exercise should be promoted. Patient compliance is crucial to determine the efficacy of the intervention.

## 1. Introduction

Low back pain (LBP) is an extremely common condition with a lifetime prevalence reported to be as high as about 80% [1]. About 5–10% of patients develop chronic pain (cLBP), responsible for high direct (health care) and indirect (lost production and lost household productivity) costs and high individual suffering and disability [2,3,4].

In particular LBP, globally, was ranked as the greatest contributor to global disability (measured in years lived with disability—YLD), and the sixth in terms of overall burden (measured in Disability-Adjusted Life Year—DALY) [5].

Low back pain is commonly classified as non-specific (90%) or specific (10%) as to reported cause and as acute (<6 weeks), subacute (6–12 weeks) or chronic (more than 12 weeks) according to duration of symptoms [1].

LBP is a multifactorial condition that develops as result of the interaction of several risk factors: constitutional risk factors (genetic predisposition, persons ages 40–80 years, and female sex), occupational risk factors (excessive static or dynamic loading, high number of lifts at work, vibrations, repeated torsion and bending movements, incorrect postures), behavioral and environmental factors (smoking, obesity, sedentary lifestyle), and psychosocial factors (stress, anxiety, depression, and work dissatisfaction) [4,6].

Management of cLBP is often multidisciplinary, involving a combination of treatments like self-care (including remaining active), pharmacotherapy (NSAID, opioids, muscle relaxants, glucocorticoid, etc.), physiotherapy, cognitive behavioral therapy, massage, physical therapy (laser therapy, TENS, T.E.CA.R.^®^ therapy), spinal manipulation, acupuncture, and in some cases, invasive interventions such as glucocorticoid injections and surgical procedures [4]. The use of kinesiotaping could be proposed beside manipulation and exercise considering its possible positive effects on posture and pain [7,8].

Core stability has reached a wide spread in recent years, considering that several studies have observed in cLBP delayed or decreased activation of lumbar multifidi and transversus abdominus and loss of physiologic tonic activation of transversus abdominus during gait and extremity movement. Dysfunction of these muscles may determine loss of lumbar spine support, increased stress and load on the joints and ligaments of lumbar spine [9,10,11,12,13,14].

The purpose of core stability exercises is to recreate normal muscle function in order to increase spinal stability, neuromuscular control within the lumbopelvic region, induce inter-segmental stiffness and prevent shear force that causes injury to the lumbar spine [15,16,17].

The aim of the present study was to review the available evidence about the effectiveness of core stability for non-specific cLBP.

## 2. Materials and Methods

We perform a systematic research on common Medline databases: PubMed, Pedro, and Cochrane Library.

Studies were selected using the following keywords: “core stability” or “core stabilization” or “lumbar stabilization” or “core strengthening” combined with the term “low back pain”, searching.

Search results were limited to studies written in English and published between January 2005 and November 2020.

Selection criteria:

We included studies that met the following inclusion criteria:-Articles published in English,-Study population aged between 18 years and 80 years,-Randomized controlled trial, meta-analysis, and systematic review on core stability in the management of non-specific chronic low back pain.

We excluded: -Acute and subacute LBP;-LBP with specific etiologies;-LBP in pregnant women, athletes, and military personnel.

Articles were screened by title and abstract. Studies that resulted unclear from their title or abstract were reviewed according the selection criteria through full-text. Furthermore, we searched reference lists of all included studies to identify other potentially relevant studies.

After the initial research on the different electronic databases, we excluded duplicate articles. The search provided a total of 420 articles. Forty-nine articles met the inclusion criteria and 371 articles were excluded (Table 1).

## 3. Results

### 3.1. Effect of Core Stability on Short Duration (3–12 Months) and Long Duration (More Than 1 Year) cLBP

Kumar et al. evaluated the effect of core stability on cLBP patients with short pain duration (3–12 months) and long pain duration (more than 1 year) and concluded that core stability is an effective rehabilitation strategy for all cLBP patients regardless the duration of symptoms, improving pain, disability, and activation of Transversus Abdominis and gluteus maximus strength [18].

### 3.2. Progressive vs. Conventional Core Stability

A recent RCT, compared the short-term effects of two core stability interventions (progressive vs. conventional) and confirmed that core stability improves pain severity, functional disability, trunk endurance, lumbopelvic control, and body balance with no significant differences between the two interventions [19].

### 3.3. Core Stability vs. Minimal or No Intervention 

Nine of forty-nine selected articles compare core stability with minimal or no intervention [20,21,22,23,24,25,26,27,28].

In particular, the effects on the following outcomes were compared: pain [18,19,20,22,23,24], disability [21,22,23,25,26], quality of life [21,25], muscle thickness [20,21,26,27,28], and muscle strength [22,24,25].

In each study, core stability was effective in improving the outcomes evaluated.

In three articles, the reduction of pain was statistically greater in the core stability group, compared with minimal or no intervention [20,21,24]. In four articles, core stability was more effective in reducing disability [21,22,23,26].

In all studies, core stability was more effective in improving quality of life [21,25], muscular thickness (in particular transversus abdominus, multifidus, obliquus abdominis, and gluteus maximus) [20,21,26,27,28], and lumbar muscular strength [22,24,25].

Two studies evaluate the circulatory (induced tissue blood flow) and biochemical effects (Plasma β-endorphin level) of core stability, always compared with minimal or no intervention. The findings indicate that the mechanism of action of the pain-relieving effect of core stability might be related to a plasma β-endorphin elevation mechanism and tissue blood flow improvement in the pathological area as part of its effects [29,30].

### 3.4. Core Stability vs. General Typical Strengthening and Stretching Training

One review and fourteen articles compared the effect of core stability with general strengthening superficial muscle training, 1 compared core stability with stretching exercises, while 1 explored the effectiveness of lumbar stabilization exercise program in addition to general strengthening and stretching exercises.

Wang et al. reviewed the effects of core stability exercise or general exercise for patients with non-specific cLBP. They selected a total of 5 trials involving 414 participants. The results of their meta-analysis indicated that core stability exercise was superior to general exercise in pain improvement (VAS/NRS) and back functional status (Oswestry Disability Index) in the short term, with no significant differences in pain relief in the intermediate- (6 months) and long-term (12 months) follow-up periods [31].

Others authors [30,31,32,33,34,35,36,37,38,39] compared core stability with typical strengthening training and confirmed that both exercises effectively reduce pain and functional disability in individuals with cLBP, but improvement was superior in the core stability group.

Although the above-mentioned studies have shown greater effectiveness of core stability than routine exercises, particularly in the short term, other studies indicate that both types of exercises have similar effects [40,41,42,43,44,45].

França et al. compared the efficacy of core stability and stretching exercises in patients with cLBP. Both treatments were effective in relieving pain and in decreasing functional disability, while core stability had more significant improvements with in addition to an improved Transversus Abdominis muscle activation [46].

Stankovic et al. evaluated the effectiveness of a lumbar stabilization exercise program in combination with general strengthening and stretching exercises and showed that a combined exercise program is more effective in reducing pain, improving disability, and quality of life than the traditional exercises alone [47].

### 3.5. Core Stability vs. Pilates, Cognitive Functional Treatment, or Proprioceptive Neuromuscular Facilitation

Four articles compared core stability with Pilates, cognitive functional treatment, or proprioceptive neuromuscular facilitation.

Bhadauria et al. concluded that significant improvements are achieved in core stability and Pilates group, while the improvement was significantly greater with the lumbar stabilization program for all the outcome measures (pain, ROM, disability, and core strength) [41]. Akodu et al. found both approaches effective without significant differences in terms of outcome measures [48].

Two studies concluded that cognitive functional treatment, proprioceptive neuromuscular facilitation, and lumbar stabilization treatment improve lumbar movement control and pain compared with the control group, with no significant difference between the experimental groups [49,50].

### 3.6. Core Stability vs. Treadmill Walking Exercise

Lumbar stabilization exercises showed superiority to treadmill walking exercises in activating the multifidus muscle, reducing pain intensity and disability in cLBP patients [51].

### 3.7. Core Stability Using Training Device vs. on a Mat

Five studies evaluated the effects of core stability exercises using equipment such as ball, flexi-bar, unstable surface, sling, or device for assisted sit-up, compared to general core stability exercises on a mat.

In particular, performing core stability exercises on an unstable surface [52] or using flexi-bar [53] should be beneficial for improving lumbar pain, muscle strength, stability, disability, and depression. In addition, performing stabilization exercises with a ball allows a greater increase in the multifidus cross-sectional areas [54], while no statistically significant differences were found performing sling core stability exercises or assisted sit-up exercises using a new training device (HubEX-LEX^®^) and mat exercise program [55,56].

### 3.8. Combination of Core Stability and Other Exercises 

Greater prevalence of diaphragm fatigue was found in cLBP individuals compared with healthy controls, which results in lack of active spinal control [57]. Four articles investigated the effect of core stability in combination with respiratory training. In particular, the association of lumbar stabilization exercises with respiratory training is more effective than lumbar stabilization alone, improving respiratory variables (maximum inspiratory pressure and maximum voluntary ventilation) by enhancing posture and stability [58,59], disability, and stabilizer muscles thickness such as diaphragm, transversus abdominis, and multifidus [59,60].

Additionally, inspiratory training (using respirometer) with lumbar stabilization exercise proved to be superior in terms of improving pain compared to expiratory training (using ball and balloon) with lumbar stabilization exercises [61].

Core stability increases contracted thickness and activity of gluteus maximus [18,26], which plays an important role in stabilizing the pelvis and is involved in patients with LBP [62]. Performing lumbar stabilization exercises with specific exercises for the gluteus muscle is more effective to increase the lumbar low back pain disability index, isometric muscle strength of lumbar flexion and extension, and stability [63].

You et al. showed that adding ankle dorsiflexion to “drawing in” the abdominal wall results in improved benefit on physical disability, pain, and core instability [64].

### 3.9. Core Stability Plus Other Classic Chronic Non-Specific LBP Treatments

Core stability exercises combined with other chronic non-specific LBP treatments have shown greater effectiveness.

In particular, the association with Neuromuscular electrical stimulation (NMES) resulted in greater analgesia, improved function, disability, and lumbopelvic stability in patients with non-specific cLBP [65], combined with thoracic spine mobilization resulted in a significant increases in the strengths of the trunk flexor and extensor muscles [66], while combined with myofascial release technique was more effective in terms of a greater increase in core stability endurance and spinal mobility (in the sagittal plane) [67].

## 4. Discussion

LBP is one of the most common pathology worldwide. Non-specific chronic low back pain is defined as low back pain without underlying specific cause and symptoms lasting over 12 weeks.

cLBP should be approached with a comprehensive treatment strategy, considering pharmacological, psychological (cognitive behavioral therapy (CBT), progressive relaxation, and biofeedback), physical, and rehabilitation strategies, and eventually minimally invasive or invasive approaches [68].

Rehabilitation treatments can involve different techniques ranging from spinal manipulations, mobilization, advice, general exercises, and specifically tailored exercise. 

Lack of stability of the spine seems to have a key role in the development of LBP and it is arguable that therapeutic exercise aimed to retrain motor skills and the activation of local spinal stabilization muscles should be proposed in a multidisciplinary approach. Currently, the use of core stability in clinical practice is growing, accompanied by a growing number of level I and level II studies conducted in the last years.

Increasing evidence supports core stability exercises in comparison to no intervention, shame intervention, or rest in improving pain and disability, strengthening that rest in LBP should be avoided and exercise should be promoted.

When core stability is compared to general exercise protocols, most of the studies observed the superiority of core stability, while only four studies evidenced similar effects.

However, the combination of core stability with other exercise modalities seems to lead to a greater improvement in pain and disability compared to both treatments alone.

Furthermore, the combination of core stability with respiratory training (in particular inspiratory training), specific exercises for the gluteus muscle, NMES, thoracic spine mobilization or myofascial release technique allows a greater improvement. Respiratory resistance determines strong contractions of the abdominal muscles and diaphragm with an increase in the intra-abdominal pressure, contributing to a decrease in lordotic curve, promoting postural adjustment.

Hip extensor (gluteus maximus) and abductor (gluteus medium) should also be evaluated, considering that lack of strength in gluteal muscles has been linked to LBP. Gluteal muscles are crucial to modulate forces between lower limbs to the spine and impaired hip extensor function may cause increased L5-S1 and sacroiliac joint pressure, which lead to functional failure and low back pain. Different studies have revealed hip muscle impairment especially in female population, that could be targeted with combined programs.

The use of unstable surface, flexi-bar, and balls could be proposed, considering that these approaches have been linked with improvement on pain and disability. Balance strategies (as unstable surfaces and ball exercises) lead to improvements in motor control, activation of multifidus, and muscle strength.

The success of core stability exercise interventions is dependent on the high adherence of the patients and the correct dosage. Even non-conclusive considerations should be achieved, as a recent review examined the dose–response-relationship of stabilization exercise interventions in non-specific cLBP patients recommending 20–30 min time per session (Grade A), three to five times a week (Grade C), while no impact of the duration of intervention in weeks on the pain intensity was demonstrated [69].

## 5. Conclusions

Core stability may provide great therapeutic effects in patients with non-specific chronic low back pain, reducing pain intensity, functional disability, and improving quality of life, core muscle activation, and thickness.

Our review aimed to better elucidate the current evidences on the role of core-stability considering only high quality studies and grouping the studies on the basis of intervention modalities.

Several studies indicate that core stability exercises are certainly more effective than rest or minimal intervention while conflicting evidences are present about the superiority of core stability exercise in comparison other types of exercise for chronic LBP.

However, the combination of core stability with other exercise modalities seems to lead to a greater improvement in pain and disability compared to treatments alone.

The success of the core stability program depends on the patient’s compliance and the correct dosage of the exercise program, which should be customized for each patient.

## Figures and Tables

**Table 1 jfmk-06-00037-t001:** Characteristics of included studies.

Article	Patient Characteristic, Sample Size, Sessions	GROUP A	GROUP B/C	Outcomes	Follow Up	Limits
18. Kumar et al., 2015;	Aged = 20–40; *n* = 30 3 days/week for 6 weeks; warm up, flexibility, and core muscle stability exercises	LBP more than 12 months	LBP between 3 and 12 months	Pain (NPRS), disability (ODI), back endurance (Sorensen test), lumbar flexibility (Modified-Modified Schober’s test), Gluteus Maximus strength (Jamar Hydraulic Hand Dynamometer) and activation of Transversus abdominis (pressure biofeedback unit)	6 weeks	No control group; the flexibility and endurance also depend on the other factors like age, gender, weight, lumbar lordosis, etc. but it was not analyzed in present study; no long-term follow-up
19. Chan et al., 2020;	Aged = 18–42; *n* = 30. 3 exercises, 3 sets, 10 repetitions with 5 s contraction hold	Progressive DMST core stability. Four stages which progressively increased in intensity on weekly basis	conventional MGB3 core stability.	Pain (NPRS), disability (RMDQ), trunk endurance (trunk flexion, trunk extension, lateral musculature), lumbopelvic control (Sahrmann 5-level core stability test), and body balance (Y-balance test)	3 weeks, 6 weeks	No control group; does not exclude patients with a history of lower limb injuries; no long-term follow-up
20. Paungmali et al., 2017;	Aged = 19–47 *n* = 25 Every participant performed 15 min of 3 different exercises in a randomized manner, with 48 h between sessions	Lumbopelvic core stabilization exercises in the supine crook lying position with the hip and knee in 70° and 90° of flexion, respectively. pressure biofeedback unit was inflated to 40 mmHg.	PLACEBO INTERVENTION = Automated passive cycling activity (30 rotations per minute) performed in the same supine crook lying position. CONTROLLED INTERVENTION = rest in the supine crook lying position	Pain (VAS), thermal pain threshold (thermal sensory analysis), pressure pain threshold (pressure algometer)	Post-intervention	The study considered only the immediate effects of LPST on pain; did not account for the different types of subgroups in chronic low back pain and several factors, such as sleep and psychosocial factors, that may influence sensory perception and sensory testing; subjective nature of the VAS pain tool
21. Noormohammadpour et al., 2018;	Aged = 18–55 *n* = 36 8 weeks of training, two floor exercises and two exercises with a Swiss ball. Each exercise was performed for three sets (morning, mid-day, and night) with ten repetitions and a ten-second holding position in each repetition	Core stability exercises based on a progressive pattern over time	Control group was kept on a waiting list and did not receive any instruction about an exercise program	Pain (VAS), disability (RDQ), quality of life (SF-36), diameter of lateral abdominal muscles (US assessment)	8 weeks	Small sample size; notable loss to follow-up; short duration of the intervention; All study participants were females; The ultrasonography assessment was only conducted in the hooklying position; the control group in our study received no intervention
22. Abass et al., 2020;	Aged = 20–60 *n* = 40 3 days/week for 8 weeks;	Lumbar stabilization exercises (three phases) in addition to conventional therapy (transcutaneous electrical nerve stimulator and infrared)	Conventional therapy (transcutaneous electrical nerve stimulator and infrared)	Pain intensity (VAS), disability index (RODQ), kinesiophobia level (TSK questionnaire) and back muscle endurance (prone double straight-leg raise test)	8 weeks	Lack of randomization in assigning participants into groups
23. Cho et al., 2015;	Aged = 37–55 *N* = 30 3 times a week for 6 weeks	Lumbar stabilization exercise program consisting of stretching as a warm-up (5 min), lumbar stabilization exercises (30 min), and stretching as a cool-down (5 min).	Hot pack (20 min), interferential current therapy (15 min), and ultrasound (5 min) for 40 min per session	Disability (ODI), lumbar lordosis angles (plain radiography)	6 weeks	Small sample size; short duration of intervention and follow up.
24. Ko et al., 2018;	Aged = 30–40 *N* = 29 60 min, three times a week for 12 weeks. The program included 10 min of warm-up, 40 min of main exercise, and 10 min of cool-down.	INTERVENTION 1 = lumbar stabilization exercise group INTERVENTION 2 = sling exercise group	CONTROL GROUP	Pain (VAS), lumbosacral region angle (X-ray images of the lateral view of the lumbar region), lumbar muscle strength (isokinetic muscle strength analyzer) and flexibility (flexibility test)	12 weeks	Small sample size; all study participants were females
25. Alp et al., 2014;	Aged = 30–40 *N* = 48 6 weeks, 3 times/week, 60 min/day	Core stabilization exercise; warming (5 min), stretching (5 min), stabilization exercises for the multifidus/transversus abdominis muscles (30 min), and cooling (5 min),	Conventional exercise	Pain (VAS), disability (RDQ), quality of life (SF-36), abdominal and back endurance (Sorensen test and Kraus-Weber test), functional ability (timed sit to stand test).	12 weeks	All study participants were females; short follow-up period
26. Narouei et al., 2020;	Aged = 18–45 *N* = 32 5 days per week for 4 weeks	16 core stabilization exercises	Control group (transcutaneous electrical nerve stimulation and a ‘hot-pack’)	Pain (VAS), disability (ODI), maximum bilateral activity of transversus abdominis, multifidus and gluteus maximus muscles (EMG), rest and contracted thickness of these muscles (US imaging)	4 weeks	Short follow-up period, use of skin-surface electrodes instead of intramuscular fine wire electrodes
27. Leonard et al., 2015.	Aged = 18–50 *N* = 25 The interventions were carried out by randomization with 48 h between the sessions	Lumbo-pelvic core stabilization training	PLACEBO TREATMENT = passive cycling CONTROLLED INTERVENTION = rest	Thickness of TrA at rest and during contraction (real time US)	Post-intervention	Small sample size; the study considered only the immediate effects of LPST on pain;
28. Gong 2016.	Aged = 20–40 *N* = 30	TRAINING GROUP = three sets of running in place in a limited area with abdominal drawing-in maneuvers each time, three times a week for six weeks	CONTROL GROUP = maintained daily living without any particular exercise	External obliquus abdominis, internal obliquus abdominis, transversus abdominis thicknesses (US imaging)	6 weeks	Small sample size
29. Paungmali et al., 2018.	Aged = 19–48 *N* = 24 All participants performed each type of exercise for approximately 15 min randomly with 48 h between sessions	Lumbar core stabilization exercise. Supine crook lying position with the hip and knee in 70° and 90° of flexion, respectively. Pressure biofeedback unit was inflated to 40 mmHg	PLACEBO TREATMENT = passive cycling in crook lying using automatic cycler) CONTROLLED INTERVENTION = positioning in crook lying and rest	Plasma β-endorphin and plasma cortisol (enzyme-linked immunosorbent assay and electrochemiluminescence)	Post-intervention	Did not directly investigate the changes in pain intensity in relationship to changes in the levels of PB and PC during LCSE; small sample size; not consider long-term follow-up of PB and PC levels.
30. Paungmali et al., 2016	Aged = 19–48 *N* = 25 All participants performed each type of exercise for approximately 15 min randomly with 48 h between sessions	Lumbopelvic stabilization training supine crook lying position with the hip and knee in 70° and 90° of flexion, respectively. pressure biofeedback unit was inflated to 40 mmHg	PLACEBO TREATMENT = passive cycling in crook lying using automatic cycler) CONTROLLED INTERVENTION = positioning in crook lying and rest	Lumbopelvic stability (pressure biofeedback device), tissue blood flow (laser Doppler flow meter)	Post-intervention	The study considered only the immediate effects of LPST on tissue blood flow and lumbopelvic stability
31. Wang et al., 2012; REVIEW	5 RCT involving 414 participants (over 18 years of age)	Core stability exercise training	Control group (general exercise)	Pain intensity (VAS, NRS, Mcgill), disability (ODI, RDQ).		Relatively low quality data that had a high risk of bias, small sample size, numerous articles did not contain sufficient information for evaluating the quality and clinical relevance of the data
32. Akhtar et al., 2017.	Aged = 39–53 *N* = 120 40min/session, one time/week for 6 weeks. All the subjects were managed with the base line treatment of therapeutic ultrasound and TENS at lumbar spine	Core stabilization exercise	Routine physical therapy exercise	Pain (VAS)	2 weeks, 4 weeks and 6 weeks	
33. Akbari et al., 2008	Aged = 36–44 *N* = 49 8 weeks, twice per week, 30 min per session	Core stabilization exercise	General exercise group	Lumbar multifidus and transversus abdominis muscles thickness (US imaging), pain (VAS) and activity limitation (Back Performance Scale).	8 weeks	
34. Waseem et al., 2019	Aged = 39–53 *N* = 120 All the subjects were managed with the base line treatment of therapeutic ultrasound and TENS at lumbar spine	Core stabilization exercise	Routine physical therapy exercise	Disability (ODI)	2 weeks, 4 weeks, 6 weeks	No proof of patients’ compliance with exercises performed at home; not accounted the type of job
35. Andrusaitis et al., 2011	Aged = 30–55 *N* = 15 40-minute physical therapy session three times a week for a total of 20 sessions. All the sessions began with a ten-minute warm-up on an ergometric bicycle	GROUP A = core stabilization exercise (stabilization exercises were taught, starting with the dorsal decubitus and progressing to the ventral decubitus, in seated, four-support, and standing positions. Increases in the number of exercises performed in each session (or load progression) occurred according to individual tolerance)	GROUP B = Core strengthening exercise (strengthening the abdominal, back, and hip muscles with an average of three series of ten repetitions of each exercise. Increases in the number of exercises performed in each session (or load progression) occurred according to individual tolerance) GROUP C = control	Pain (VAS), disability (ODI), balance tests (Balance Master^®^ System)	7 weeks	Small number of subjects and the differences in the duration of cLBP between the groups.
36. França et al., 2010	Aged = 34–50 *N* = 30 30min/session, twice/week, 6 weeks.	Core stabilization (exercises focused on the TrA and lumbar multifidus muscles)	Superficial strengthening (exercises focused on the rectus abdominis, abdominus obliquus internus, abdominus obliquus externus, and erector spinae)	Pain (VAS, McGill pain questionnaire), disability (ODI), and TrA muscle activation capacity (Pressure Biofeedback Unit).	6 weeks	No intermediate and long-term follow up examinations. Biopsychosocial factors were not observed in this study.
37. Gatti et al., 2011	Aged = 45–71 *N* = 79 The intervention consisted of 2 sessions per week, each lasting 60 min (15 min of walking on a treadmill, 30 min of flexibility exercises and 15 min of trunk balance exercises or strengthening exercises for the limbs and trunk), for a total of 10 treatments over 5 consecutive weeks.	Trunk balance exercises in addition to trunk flexibility exercises	Strengthening exercises in addition to the same standard trunk flexibility exercises	Pain intensity (VAS), disability (RDQ), quality of life (SF-12), painful positions, use of analgesic drugs, and referred pain	5 weeks	Lack of an a priori sample size analysis based on the primary outcomes. Absence of a follow-up beyond the termination of the intervention period. Placebo or Hawthorne effect cannot be excluded, as it was not possible to blind the patients to the intervention,
38. Kwon et al., 2020	Aged = 22–40 *N* = 30 3 times per week for 6 weeks	Lumbar stabilization exercises in different postures,	Ordinary trunk muscle strengthening exercise	Pain intensity (VAS), transversus abdominis activation capacity (pressure biofeedback unit), transversus abdominis thickness (US imaging) and disability(K-ODI).	6 weeks	Small sample size; not accounted multifidus and pelvic floor muscles
39. Sipaviciene et al., 2020	Aged = 32–44 *N* = 70 20-week exercise programs	Lumbar stabilization exercise program. strengthen the deep trunk stabilizing muscles (especially transverse abdominal, internal oblique and lumbar multifidus) and control pelvic muscles. 8–16 repetitions of all exercises	Lumbar muscle strengthening exercise program. exercises that improve trunk flexor (rectus abdominis) and extensor (erector spinae) muscles strength. 8–16 repetitions of all exercises	Pain (VAS), disability (ODI), cross-sectional area of the multifidus muscle (US imaging), isokinetic peak torque (isokinetic dynamometer)	20 weeks, 24 weeks, 28 weeks, 32 weeks	Do not investigate the long-term effects of a lumbar stabilization exercise program
40. Inani et al., 2013	Aged = 20–50 *N* = 30 12 week exercise program	Core stabilization exercises (4 phases)	Conventional exercises	Pain (VAS), disability (modified ODI),	3 months	Small sample size, empirical verification of transversus abdominis and multifidus muscle contraction and recruitment
41. Bhadauria et al., 2017	Aged = 20–47 *N* = 44 Ten sessions of exercises for 3 weeks were prescribed along with interferential current and hot moist pack.Warm up stretching exercises for 10 min before the main exercises and cool down exercises for 5 min after each session.	GROUP A: lumbar stabilization group (16 lumbar stabilization exercises)	GROUP B: dynamic strengthening group (14 exercises, which activated the extensor (erector spinae) and flexor (rectus abdominis) muscle groups GROUP C: Pilates group	Pain (VAS), disability (ODI), range of motion-lumbar flexion and extension (modified Schober test) and core strength (pressure biofeedback unit)	3 weeks	Small sample size. Only ten exercise sessions.
42. Moon et al., 2013	Aged = 23–33 *N* = 21 Exercises were performed for 1 h, twice weekly, for 8 weeks	Lumbar stabilization exercise group	Lumbar dynamic strengthening exercise group	Pain (VAS), disability (ODI), strength of the lumbar extensors (Using MedX)	8 weeks	Small sample size, young age, short follow-up period
43. Shamsi et al., 2016	Aged = 20–60 *N* = 43 16 sessions program, 3 times/week Warm up 5 min, The pure exercise time for core stability group was 20 min and for general exercises was 14 min in each session	Core stability exercises	General exercises	Pain (VAS), disability (ODI), Endurance core stability tests (trunk flexor; (2) trunk extensor; and (3) side bridge tests)	5 weeks	The study design was a quasi-randomized controlled trial. The mean age was higher in the GE than the CSE group. Short follow up period
44. Shamsi et al., 2020	Aged = 20–60 *N* = 56 16 sessions program, 3 times/weekWarm up (8 stretching exercises and stationary cycling for 5 min), The pure exercise time for core stability group was 20 min and for general exercises was 14 min in each session	Core stability exercises	General exercises	Pain (VAS), disability (ODI), trunk muscle activation patterns (EMG)	5 weeks	Lack of a true control group, lack of blindness for the treating physiotherapist, quasi-randomized trial design.
45. Nabavi et al., 2018.	Aged = 20–50 *N* = 41 1-h treatment, 3 times/week for 4 weeks. Both groups received routine physiotherapy including electrotherapy and warmup exercises	Core stability exercises	General exercises	Pain (VAS), muscle dimensions of transverse abdominis and lumbar multifidus muscles (US imaging)	4 weeks	No long-term follow up, lack of blindness for the treating physiotherapist
46. França et al., 2012	Aged = 30–50 *N* = 30 6 weeks, twice per week, lasting 30 min each	Segmental stabilization exercises (exercises focused on the TrA and LM muscles according to the protocol proposed by Richardson)	Stretching of trunk and hamstrings muscles (stretching of erector spinae, hamstring, and triceps surae muscles and connective tissues posterior to column)	Pain (VAS and McGill pain questionnaire), disability (ODI) and TrA muscle activation (Pressure Biofeedback Unit)	6 weeks	Lack of a true control group (no treatment), no long-term follow up, small sample size. A more specific analysis of the LM and TrA muscles using ultrasound imaging or electromyography was not performed.
47. Stankovic et al., 2012	Aged = 40–60 *N* = 160 20 therapeutic treatments, for 4 weeks (5 days per week). Each treatment lasted 30 min.	Lumbar stabilization exercises (15 exercises)	Strengthening and stretching of the large, superficial back muscles	Pain (VAS), Disability (ODI), quality of life (SF-36)	4 weeks	Subjective tests
48. Akodu et al., 2016	Aged = 35–60 *N* = 29 twice weekly for 4 consecutive weeks.	GROUP 1: core stabilization exercise + infra-red radiation,	GROUP 2: Pilates exercise + infra- red radiation GROUP 3: infra-red radiation and back care education	Pain (NRS), disability (RMDQ), lumbar range of motion (MST), level of physical activity (IPAQ)	2 weeks, 4 weeks	
49. Khodadad et al., 2020	Aged = 40–45 *N* = 52 60 min session, 3 days per week, for 8 weeks. warm-up stretching exercises for 10 min before the main exercises, and cool-down exercises for 10 min after each session	Lumbar stabilization treatment (Five exercises that activate the deep lumbar stabilizing muscles: the transversus abdominis, lumbar multifidi, and internal obliques)	Cognitive functional treatment	Pain (VAS), lumbar movement control (Luomajoki LMC battery tests)	8 weeks	Small sample size, only male adults aged 40–50 years patients. lack of blindness for the treating physiotherapist, short follow-up period.
50. Areeudomwong et al.	Aged = 18–50 *N* = 45 three weekly 30 min sessions over four weeks	GROUP 1 = core stabilization exercise	GROUP 2 = proprioceptive neuromuscular facilitation CONTROL GROUP (5 min to 10 min of therapeutic ultrasound depending on treatment area, 20-min general trunk strengthening exercise program was performed in three sets of 10 repetitions, with a 30 s rest between repetitions and 60 s rest between sets)	Pain (NRS), disability (RMDQ), patient satisfaction superficial and deep trunk muscle activity (EMG)	4 weeks, 3 months	Only investigated effects on pain and electromyographic activity of trunk muscles; short term follow-up
51. Bello et al., 2018	Aged = 30–50 *N* = 50 three times a week for 8 weeks	Lumbar stabilization exercises following the McGill protocol (30 min of stabilization exercises per session)	Modified Bruce treadmill walking protocol	Pain (VAS), functional disability (ODI) and multifidus muscle activation (surface EMG)	8 weeks	Use of skin-surface electrodes instead of intramuscular fine wire electrodes. Unable to measure EMG activity of the multifidus during dynamic activity (movement)
52. Kang et al., 2018	Aged = 30–50 *N* = 2430 min/session, 5 times/week for 6 weeks 10-min hot pack treatment, 15-min electrotherapy and 5-min ultrasonic treatment before the lumbar stabilization exercise	Lumbar stabilization exercises performed on unstable surface	Lumbar stabilization exercises performed on stable surface	Pain (VAS), disability (ODI), Back muscle strength (digital back muscle strength meter), proprioception and lumbar spine stability (SBST), depression (BDI)	6 weeks	Patients are adults with CLBP working in an automobile assembly plant (they are not a representative sample of all patients). Not measured muscle activation level or strength of specific muscle. It is not possible to determine which exercise specifically improved certain dependent variables.
53. Chung et al., 2018	Aged = 25–40 *N* = 27 30 min/day, 3 times a week, for 6 weeks. warm up for 5 min and cool down for 5 min	Lumbar stabilization exercises using flexi-bar	Stabilization exercises	Pain (VAS), disability (ODI), TrA activation capacity (pressure biofeedback unit) and thickness (US imaging)	6 weeks	Lack of blindness for the therapist and patients. Small sample size, short follow-up period. TrA activation capacity and thickness in dynamic conditions was not measured
54. Chung et al., 2018	Aged = 25–45 *N* = 24 Warm up and cool-down: walking for 10 min each on a treadmill.three times/week, 8 weeks, using four different motions	Stabilization exercise program using balls	Stabilization exercise program (same motions on a mat)	Pain (VAS), disability (ODI), Multifidus muscle cross-sectional areas (TC), left and right weight bearing differences (Tetrax balancing scale)	8 weeks	Small sample size, relatively short intervention period (eight weeks), stability in dynamic conditions was not measured.
55. Yoo et al., 2012	Aged = 19–21 *N* = 30 3 times/week, 4 weeks	Core Stabilization Exercises Using a Sling (6 movements. Each action was maintained for 20 s followed by 10 s rest. 3 sets, 6 repetitions/set with 90 s rest between sets)	Core Stabilization Exercises on a mat (8 movements. Each action was maintained for 10 s followed by 10 s rest. 2 repetitions/sets, 2 sets with 15 s rest between sets)	Pain (VAS), extensor muscle strength (tergumed device)	4 weeks	Quality of life and disability are not accounted, small sample size, short-term follow up
56. Bae et al., 2018	Aged = 20–45 *N* = 36 30 min/session. Standard trainings such as warm-up, cool-down, and stretching were conducted both before and after the exercise program with the same method. 12 sessions of the exercise program, 4 weeks.	Assisted sit-up exercise with a training device (HubEX-LEX^®^)	Conventional core stabilization exercise	Pain (VAS), disability (ODI), abdominal muscle thickness (US imaging) and activity of core muscles (surface EMG)	4 weeks, 8 weeks, 16 weeks	Absence of evaluation for dorsal paraspinal muscles. Small sample size. Young age.
58. Mohan et al., 2020.	Aged = 20–45 *N* = 40 3 times/week, 8 weeks	Core stability with a combined ball and balloon exercise with routine physiotherapy (ultrasound, spinal flexion or extension exercises)	Routine physiotherapy (ultrasound, spinal flexion or extension exercises)	Maximum inspiratory pressure, maximum expiratory pressure, maximum voluntary ventilation (spirometer). pain (NRS), faulty breathing pattern (total faulty breathing scale), chest expansion (cloth tape measure) and lumbo-pelvic stability (pressure biofeedback device)	8 weeks	Lack of appropriate training and understanding among the participants who performed MVV maneuvers. Did not account for psychological issues which might affect respiration. there were no normative values for the variables tested in this study to compare with those in NS-LBP patients
59. Oh et al., 2020	Aged = 40–49 *N* = 44 50 min per session, 3 sessions per week for 4 weeks 5 min of stretching to warm up and cool down was performed, and 3–5 sets with each lasting for 20 s were performed for each exercise program. Between the sets, a 1-min break was allowed	Abdominal draw-in with a lumbar stabilization exercise program and respiratory resistance exercise (expand-A-Lung^®^)	Abdominal draw-in with a lumbar stabilization exercise program	Pain (QVAS), disability (ODI-K), diaphragm thickness, and contraction rate (US imaging), and lung capacity test (Microquark)	4 weeks	Hand pressure and direction of the ultrasonic probe was not the same during the measurement. Participants were all females ages 40–49 years. The abdominal contraction ability or maximal inspiratory pressure and maximal expiratory pressure were not assessed, the participants were patients admitted to a hospital, making it difficult to control the social participation, physical activity, and medical treatments, the psychological characteristics of the outcome measures could not be assessed.
60. Finta et al., 2018	Aged = 18–25 *N* = 47 8-weeks, 2 session/week, 60 min/session 10 min warm up, 40 min circuit training with five sections, and with 3 min of exercising in one section and 1 min breaks between the sections, 10 min cool down	Complex training program with diaphragm training (POWER breathe Medic Plus device twice a day at home, 30 inhalations per occasion, and with the speed of 15 inhalations/min and during strengthening exercises)	Complex training program	Pain (VAS) transversus abdominis, diaphragm, and lumbar multifidus muscle thickness (US imaging)	8 weeks	It is not possible discriminate between the increase of muscle thickness as a result of the changes of the tone and activation pattern and muscle hypertrophy, possible different compliance of the subjects. Possibility of different levels of contraction during US measurement
61. Kavya et al., 2020	Aged = 20–50 *N* = 36 3 weeks All the three groups received moist heat application for 10 min	Group A = inspiratory training with lumbar stabilization (5 min inspiratory training and 25 min lumbar stabilization exercise/session), group B = expiratory training with lumbar stabilization (5 min expiratory training and 25 min lumbar stabilization exercise/session)	Group C = lumbar stabilization exercise (25 min/session)	Pain (VAS), disability (ODI), core strength (pressure biofeedback unit)	3 weeks	Small sample size, no long term follow up
63. Jeong et al., 2015	Aged = 30–50 *N* = 40 6 weeks, Three times/week, 50 min/session	Lumbar segmental stabilization exercise plus exercise to strengthen the muscles of the gluteus group	Lumbar segmental stabilization exercise group	Disability (ODI), lumbar isometric strength (M3 (Schnell, Germany) isometric muscle strength measurement equipment)	6 weeks	Small sample size, physical activities, and environmental factors other than exercise were not completely controlled
64. You et al., 2013	Aged = 40–60 *N* = 40 8 weeks, three days/week	Drawing in the abdominal wall (hook-lying posture for 20 s × 10 sets, 60-s rest between the practice sessions, and three repetitions) plus ankle dorsiflexion (30% of maximal voluntary isometric contraction of the tibialis anterior muscle against resistance provided by elastic band for 20 s × 10 sets, a 60-s rest between the practice sessions, and three repetitions)	Drawing in the abdominal wall (hook-lying posture for 20 s × 10 sets, 60-s rest between the practice sessions, and three repetitions)	Pain (VAS, Pain Disability Index, Pain Rating Scale), Disability (ODI, RMDQ), core stability test (active straight leg raise)	8 weeks, 16 weeks	Small sample size
65. Luz et al., 2019	Aged = 18–35 *N* = 30 4 weeks, 3 session/week	GROUP A = CORE stability (4 exercises/session, in each exercise the posture was maintained for 10 s. 10 rep, 20-s intervals between each series and one-minute intervals between each exercise) plus NMES program addressing gluteus maximus, gluteus medius, rectus abdominis, and bilateral transverse abdominis (warm up- 5 Hz for five minutes. Then, Frequency of 35 Hz for 10 min and, finally, 80 Hz for another 10 min. The stimulus intensity was the maximum needed to produce a strong, visible muscular contraction without causing discomfort to the patient.)	GROUP B = core stability (four exercises per session, and in each exercise the posture was maintained for 10 s. Ten repetitions were performed with 20-s intervals between each series and one-minute intervals between each exercise) GROUP C = NMES group (warm up- 5 Hz for five minutes. Then, Frequency of 35 Hz for 10 min and, finally, 80 Hz for another 10 min. The stimulus intensity was the maximum needed to produce a strong, visible muscular contraction without causing discomfort to the patient.)	Pain (VAS), Disability (ODI, RDQ), hamstring flexibility (hamstring flexibility test- wells’ bench), evaluation of core stabilizing muscles (static trunk endurance, Sorenson Endurance, Side Bridge, and Prone Instability)	4 weeks	Small sample size, only female patients, short-term intervention and follow up, no control group, participants’ intake of analgesic drugs was not evaluated or controlled
66. Yang et al., 2015	Aged = 35–55 *N* = 20 12 weeks, 3 sessions/week, 1 h/dailyWarm up- 15 min walking on a treadmill Cool down- 3 stretching exercises	Stabilization exercises (active stabilization exercises- 9 exercises, 15 rep/set for 3 set) plus Thoracic mobilization therapy (manipulation therapy, reported by Kaltenborn, five minutes before starting the stabilization exercises)	Stabilization exercises	Range of motion of the spine (Spinal Mouse) isometric muscular strength of the lumbar deep muscles (isometric sthenometer, ISO-check)	12 weeks	
67. Ozsoy et al., 2019	Aged = 65–70 *N* = 45 3 days/week, 6 weeks	Core stabilization exercises ((60 min/session, starting with a 10-min warm-up program and ending with a 5-min cool-down. Exercises were designed from 1 set to 3 sets, from 8 to 15 repetitions and contractions from 5 s to 10 s. Rest intervals were set as 30 s between the sets and 2–3 min between the exercises) plus Myofascial Release Technique with a roller massager (3 sets (1 min rest between sets) lasting for 30 s for each myofascial track)	Core stabilization exercises (60 min/session, starting with a 10-min warm-up program and ending with a 5-min cool-down. Exercises were designed from 1 set to 3 sets, from 8 to 15 repetitions and contractions from 5 s to 10 s. Rest intervals were set as 30 s between the sets and 2–3 min between the exercises)	Pain (VAS, pain pressure threshold), low back disability (ODI), lower body flexibility (chair sit and reach test), kinesiophobia (tampa scale of kinesiophobia), core stability endurance (supine bridge test), spinal mobility (Spinal Mouse System), gait characteristics (Biodex Gait Trainer 2) and quality of life (WHOQOL-OLD)	6 weeks	Myofascial relaxation technique was performed with a roller massager, but many other methods also exist. Absence of Myofascial Release Technique alone group.

## Data Availability

The study does not report any data.

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
