# Peer review of "Efficacy of Core Stability in Non-Specific Chronic Low Back Pain"

_jfmk, 2021, doi:10.3390/jfmk6020037_

Round 1

Reviewer 1 Report

This is an original and interesting study that aims to review the available evidence on the efficacy of core stability exercises for non-specific cLBP.

Although the manuscript is well written and the data support the arguments, in my opinion, there are some issues that, if addressed, would improve the quality of this paper.

Please, report tabular structure(s) used to display results of individual studies and syntheses

Discuss principal limitations of the studies included in the review.

Discuss any limitations of the review processes used, and comment on the potential impact of each limitation

Describe any significant differences in core stability exercise in the various studies presented. Is it possible to hypothesize an exercise protocol that is more effective than others?

Author Response

Dear Reviewer, thank you for your important considerations.
1-2. As you suggested, we added a table of selected studies (see Table I), underlying also the limitations of each study.
3. We performed a systematic review considering inclusion and exclusion criteria argumented in methodology section. From our point of view, no major limitation could compromise our review process
4. Thank you for your consideration, we report the principal characteristics of each study in Table I. Unfortunately, a standard core stability protocol should not be proposed considering the heterogeneity of the studies and that correct dosage and level of proposed exercises should be customize on each patient.

Reviewer 2 Report

This is an original and interesting study that aims to review the available evidence on the efficacy of core stability exercises for non-specific cLBP.

Although the manuscript is well written and the data support the arguments, in my opinion, there are some issues that, if addressed, would improve the quality of this paper.

Please, report tabular structure(s) used to display results of individual studies and syntheses

Discuss principal limitations of the studies included in the review.

Discuss any limitations of the review processes used, and comment on the potential impact of each limitation

Describe any significant differences in core stability exercise in the various studies presented. Is it possible to hypothesize an exercise protocol that is more effective than others?

Author Response

(The authors gave the same response as above.)

Reviewer 3 Report

It is my pleasure to give my insights and highlight some recommendations and commentaries to the manuscript, considering my expertise in the field. I really appreciate this opportunity to discuss the evidence and I hope my contributions could serve to improve the final version of the study.

The manuscript titled “EFFICACY OF CORE STABILITY IN NON-SPECIFIC CHRONIC LOW BACK PAIN” deals with an important issue of Low back Pain. The purpose of this study was to review the available evidence about the effectiveness in reducing pain and improving disability of core stability exercises for non-specific management of chronic low back pain.

I believe that although the authors have carried out an interesting study and the content of the document has high potential and may be of interest to the field of study, there are some details that could be improved in order to increase its relevance, clarity, application practical and overall quality.

To help better readers understanding, please add in the introduction section a sentence regarding the effect of Kinesio taping in the management of chronic low back pain or add a little section (Core stability vs Kinesio taping application) as follow:

A Short Overview of the Effects of Kinesio Taping for Postural Spine Curvature Disorders. J. Funct. Morphol. Kinesiol. 2018, 3, 59. https://doi.org/10.3390/jfmk3040059

The Effects of Exercise and Kinesio Tape on Physical Limitations in Patients with Knee Osteoarthritis. J. Funct. Morphol. Kinesiol. 2016, 1, 355-368. https://doi.org/10.3390/jfmk1040355

In the conclusion section please highlight better the scientific/clinical relevance of your work. Please provide a clear “take-home message” of the importance of this paper in the scientific community.

Author Response

Dear reviewer, thank you for your considerations.
1. As you suggested, we add in the introduction the possible role of kinesiotaping in low back pain.
2. As you suggested we add our considerations on our present work in the conclusion “Our review aimed to better elucidate the current evidences on the role of core-stability considering only high quality studies and grouping the studies on the basis of intervention modalities”.